# Complex Formation between the Transcription Factor WRKY53 and Antioxidative Enzymes Leads to Reciprocal Inhibition

**DOI:** 10.3390/antiox13030315

**Published:** 2024-03-05

**Authors:** Ana Gabriela Andrade Galan, Jasmin Doll, Natalie Faiß, Patricia Weber, Ulrike Zentgraf

**Affiliations:** Center for Plant Molecular Biology (ZMBP), University of Tuebingen, Auf der Morgenstelle 32, 72076 Tübingen, Germany; ana.andrade@zmbp.uni-tuebingen.de (A.G.A.G.); jasmin.doll@zmbp.uni-tuebingen.de (J.D.); natalie.faiss@zmbp.uni-tuebingen.de (N.F.); pa.weber@student.uni-tuebingen.de (P.W.)

**Keywords:** catalase (CAT), ascorbate peroxidase (APX), superoxide dismutase (SOD), WRKY transcription factors, WRKY53, protein–protein interaction, zymograms, *Arabidopsis thaliana*, plant senescence

## Abstract

The transcription factor WRKY53 of the model plant *Arabidopsis thaliana* is an important regulator of leaf senescence. Its expression, activity and degradation are tightly controlled by various mechanisms and feedback loops. Hydrogen peroxide is one of the inducing agents for *WRKY53* expression, and a long-lasting intracellular increase in H_2_O_2_ content accompanies the upregulation of *WRKY53* at the onset of leaf senescence. We have identified different antioxidative enzymes, including catalases (CATs), superoxide dismutases (SODs) and ascorbate peroxidases (APXs), as protein interaction partners of WRKY53 in a WRKY53-pulldown experiment at different developmental stages. The interaction of WRKY53 with these enzymes was confirmed in vivo by bimolecular fluorescence complementation assays (BiFC) in *Arabidopsis* protoplasts and transiently transformed tobacco leaves. The interaction with WRKY53 inhibited the activity of the enzyme isoforms CAT2, CAT3, APX1, Cu/ZuSOD1 and FeSOD1 (and vice versa)*,* while the function of WRKY53 as a transcription factor was also inhibited by these complex formations. Other WRKY factors like WRKY18 or WRKY25 had no or only mild inhibitory effects on the enzyme activities, indicating that WRKY53 has a central position in this crosstalk. Taken together, we identified a new additional and unexpected feedback regulation between H_2_O_2,_ the antioxidative enzymes and the transcription factor WRKY53.

## 1. Introduction

Senescence is an integral part of plant development. Older or shaded leaves are not just sacrificed but exploited before shedding. Valuable nutrients are remobilized out of the senescing tissues for the sake of the whole plant during sequential senescence, and even for the next generation during monocarpic senescence. In the latter, carbon, mineral and nitrogen resources are transported to the developing fruits and seeds after the transition to the reproductive stage. Senescence in general, but especially onset and progression of monocarpic senescence, are highly complex processes which are driven by a multitude of molecular signals, including almost all plant hormones but also other small signaling molecules like small peptides, calcium or reactive oxygen species (ROS). Onset and progression of monocarpic senescence are accompanied by a drastic change in gene expression, implying a central role for transcription regulators like histone modifiers or chromatin remodeling factors, regulators of the DNA methylation and transcription activators and repressors [1,2,3,4,5,6,7,8,9,10]. Activity and function of all these components have to be tightly linked and integrated into a complex regulatory network (for review see [10,11,12,13,14]) to guarantee a smooth operation of senescence.

*Arabidopsis*, as a model system, has been extensively utilized to analyze the fundamental molecular mechanisms underlying senescence processes. This is due to the ability to progress from germination to fully senescent plants with siliques and mature seeds within approximately 10–12 weeks under long-day light conditions. Leaf material can easily be defined within the rosette using a color code considering the gradient along sequential senescence within the rosette to compare leaves which are in the same developmental stage. Moreover, a clear guideline on how senescence can be analyzed has been developed for *Arabidopsis* plants [15]. In *Arabidopsis*, two large groups of transcription factors in particular, the NAC and the WRKY transcription factor families, are overrepresented in the senescence transcriptome and play important roles in regulating developmental as well as stress-induced senescence [3,4,5,10,11,12,13,14]. Among more than a hundred transcription factors driving senescence, WRKY53 is just one cogwheel in the gear of a very complex regulatory network that drives the onset of senescence in conjunction with other. Beyond its expression, which is responsive to hydrogen peroxide, jasmonic or salicylic acid, it is regulated by at least 12 other transcription factors, and its degradation as well as its activity are also regulated. Direct phosphorylation by a MAP kinase kinase kinase 1 can increase its transactivation potential, whereas interaction with various partner proteins can inhibit its DNA-binding or change its promoter binding specificity (for review, see [14] and references therein). The HECT ubiquitin ligase (UPL5) can specifically bind to WRKY53 and send the protein to degradation via the 26S-proteasom. The opposite expression pattern compared to *WRKY53* itself ensures that WRKY53 protein is rapidly degraded when it should be expressed during stress conditions, thereby preventing premature induction of senescence [16].

It became evident during the last two decades that hydrogen peroxide, as well as other ROS, act as signaling molecules in senescence. ROS are by-products of aerobic metabolism in all organisms which cannot be avoided. They are formed by either partial reduction of or direct energy transfer to molecular oxygen (O_2_). These very reactive molecules can oxidize more or less all kinds of macromolecules, influencing a plethora of physiological changes, including the activity of transcription factors [17,18,19,20]. However, when ROS are present in excess, they can be harmful to the cells; therefore, ROS production and scavenging needs to be balanced ingeniously. Furthermore, in addition to many non-enzymatic ROS scavenging molecules, antioxidative enzymes like catalases (CATs), ascorbate peroxidases (APXs), and superoxide dismutases (SODs) are present in several isoforms in different cellular compartments, counteracting ROS production. A delicate regulation of the CAT and APX activities leads to a long-lasting increase of intracellular hydrogen peroxide during the onset of monocarpic senescence for 7 to 10 days. To initiate this long-term increase, *CAT2* gene expression is inhibited by the bZIP transcription factor G-Box binding factor 1 (GBF1) [21,22]. Since CAT2 protein contributes approximately 80% of the total CAT activity in leaves and has a high turnover rate [23], this rapidly leads to a depletion of catalase activity and an increase of intracellular H_2_O_2_ content. Aside from that, the APX1 enzyme is rendered sensitive against H_2_O_2_ by a mechanism that remains unknown so far, particularly during bolting and onset of monocarpic senescence [21,24]. This further contributes to increasing levels of H_2_O_2_, and thereby activates a positive feedback loop. With ongoing plant development, APX1 inhibition is overridden again and, additionally, *CAT3* expression and enzyme activity start to increase, and thus antioxidative capacity is at least partially restored. This model was confirmed in *gbf1* mutant plants. If *CAT2* downregulation is abolished in *gbf1* mutant plants, no H_2_O_2_ increase and no positive feedback through APX1 inhibition is initiated; thus, no long-term H_2_O_2_ increase can be observed in *gbf1* mutants, and senescence is delayed [22,25].

Remarkably, *WRKY53* gene expression can be driven by hydrogen peroxide, and this long-term increase coincides with the increase of *WRKY53* expression at this time point. In addition, all three *CAT* genes have been identified as direct target genes of WRKY53, creating further feedback regulations [26]. Moreover, the homolog of *WRKY53* of rapeseed also feeds back on hydrogen peroxide levels, in this case by altering transcription of *RbohD* and *RbohF* [27]. However, so far there is no indication that WRKY53 regulates *Rboh*s in *Arabidopsis*; this appears to be taken over by *WRKY55* [28]. Here, we characterized a new direct feedback loop involving WRKY53 and the antioxidative enzymes. While analyzing the different in vivo protein interaction partners of WRKY53 using a pulldown of WRKY53 proteins at different developmental stages, combined with LC/MS-MS, we found, beyond others, many different antioxidative enzymes including CATs, SODs and APXs. We could confirm the direct interaction between WRKY53 and these enzymes in vivo by BiFC in transiently transformed *Arabidopsis* protoplasts as well as *N. benthamiana* leaves. Interaction with WRKY53 inhibits the enzyme function of the different enzyme isoforms to different extends, and this inhibition can be realized over a wide range of plant developmental stages. Vice versa, the function of WRKY53 as transcriptional regulator is also inhibited to different extents by these interactions.

## 2. Materials and Methods

### 2.1. Plant Cultivation

*Arabidopsis* plants (*A. thaliana* Ecotype Columbia) were grown under long-day conditions (16h light) on standard soil. An amount of 70 L of the standard soil CL Topf (Art.Nr.: 10-00300, PATZER ERDEN GmbH, Sinntal, Germany) was mixed with 8 L of sand (Flammer Bauunternehmung GmbH & Co. KG, Rheinsand, Tuebingen, Germany) and sieved with a mesh width of 8 × 10 mm. Detailed soil composition is provided in Appendix A. For all experiments, only moderate light intensity (80–100 μmol s^−1^ m^−2^) in a climatic chamber at an ambient temperature of 20 °C was applied. As catalases show circadian regulation, plant material was harvested always at the same time of day to avoid circadian effects. The positions of the individual leaves within the rosette were color-coded according to their age using colored threats [15]. In all experiments, *A. thaliana* Ecotype Columbia-0 (Col-0), *upl5* or *wrky53* mutant plants in Col-0 background (SALK_116446; SALK_034157) were used. Tobacco plants (*N. benthamiana*) were cultivated for 4–5 weeks in the greenhouse on standard soil under long-day conditions under normal light intensity (120–150 μmol s^−1^ m^−2^).

### 2.2. Zymograms for Antioxidative Enzymes

For the analysis of catalase activities, we consistently utilized leaves from the same positions within the rosette (Leaf No. 5 and No. 6). Leaves from three 4- to 8-week-old *Arabidopsis* plants were ground in a solution containing 100 mM Tris-HCl pH 8.0, 20% glycerol, and 30 mM dithiothreitol (DTT). For the analyses of ascorbate peroxidase and superoxide dismutase activities, leaves No. 5–8 from five 4- to 8-week-old *Arabidopsis* plants were ground in 50 mM potassium phosphate buffer, pH 7.8, containing 5 mM ascorbate, 2% Triton X-100, 10% glycerol, and 0.25 mM EDTA. These crude extracts were then centrifuged for 30 min at 13,000 rpm at 4 °C, and the protein concentrations of the supernatants were measured using the Bradford method [29]. The resulting protein extracts were used directly for the zymograms.

To analyze the different isoforms of catalase, 10 µg of the protein extracts were separated in 6% native polyacrylamide gels (0.375 M Tris-HCl, pH 6.8, as gel buffer) for 1 h (120 V) using 250 mM glycine and 25 mM Tris-HCl, pH 8.3, as the electrophoresis buffer. After electrophoresis, the gels were stained for CAT activity, as described in [30]. The gels were soaked in 0.01% H_2_O_2_ solution for 2 min, followed by washing twice in water and incubating them for 2–5 min in 1% of both FeCl_3_ and K_3_[Fe(CN)_6_]. Rinsing the gels twice in water stopped the reaction. For immunodetection of catalases, the native polyacrylamide gels were blotted on a nitrocellulose membrane. Subsequently, the membrane was rinsed twice in Tris-buffered saline (TBS) and blocked with 3% milk powder in TBS-Tween 20 (TBS-T). Polyclonal anti-rye-CAT antibodies in 1.5% milk powder were used, followed by secondary peroxidase-conjugated antibodies for visualization.

The activity of the superoxide dismutase isoforms was analyzed using 120 µg of the protein extracts, which were separated in 13% native polyacrylamide gels (0.375 M Tris-HCl, pH 8.8, containing 10% glycerol as gel buffer). In this case, a 5% stacking gel was necessary (0.125 M Tris-HCl, pH 6.8, containing 10% glycerol as gel buffer). Electrophoresis was performed for 1 h (120 V) using 250 mM glycine and 25 mM Tris-HCl, pH 8.3, as electrophoresis buffer. Afterwards, gels were washed in water and rinsed in staining solution (50 mM potassium phosphate buffer, pH 7.0, 1 mM EDTA, 5.2 µm Riboflavin and 0.5 mM NBT) for 30 min while shaking in the dark. Subsequently, the gels were washed twice in water and illuminated for 30 min, as described in [31].

For the analyses of the ascorbate peroxidase isoforms, 40 µg of the protein extracts were separated on the same gels as for the superoxide dismutase isoforms using 250 mM glycine and 25 mM Tris-HCl, pH 8.3, containing 2 mM ascorbate as an electrophoresis buffer. After electrophoresis, the gels were soaked 3 times in 50 mM potassium phosphate buffer, pH 7.0, containing 2 mM ascorbate for 10 min, subsequently, in 50 mM potassium phosphate buffer, pH 7.0, containing 4 mM ascorbate and 1 mM H_2_O_2_ for 20 min. After washing in water, 50 mM potassium phosphate buffer, pH 7.8, containing 14 mM TEMED (N,N,N′,N′-tetramethylethylenediamine) and 2.45 mM NBT (nitro blue tetrazolium) was used to stain the gels for 10–30 min [32].

### 2.3. Transient Transformation of Arabidopsis Protoplasts and Arabidopsis or Tobacco Leaves

*A. thaliana* ecotype Col-0 root cells that were grown in liquid cultures were used to prepare protoplasts and transform them transiently with plasmid DNA. For protoplast preparation and transient transformation using PEG, we followed the protocol described in [33]. *N. benthamiana* plants were infiltrated with *Agrobacterium tumefaciens* suspension cultures containing the BiFC constructs. LB media, with the respective antibiotics, was inoculated with an overnight culture of the bacteria and incubated for 4–6 h. Subsequently, this culture was centrifuged at 4000 rpm for 10 min. The bacterial pellet was diluted in infiltration media (10 mM MgCl_2_, 0.5 M MES, 100 mM Acetosyringone) to an OD_600_ of 0.5. Leaves of 4-week-old tobacco plants were infiltrated by manual injection using a 1-mL needleless syringe. *A. thaliana* leaves of 3- to 4-week-old plants were co-infiltrated using the same procedure with *A. tumefaciens* suspension cultures containing the GUS-reporter and the effector construct, respectively.

### 2.4. Bimolecular Fluorescence Complementation (BiFC), Cytometry, and Confocal Microscopy

In order to study the interaction of WRKY53 with the antioxidative enzymes CAT2, CAT3, APX1, and Cu/ZnSOD1, ratiometric BiFC assays were performed. Therefore, the cDNAs of the subunits of *CAT2* (2393 bp, At4g35090) and *CAT3* (2784 bp, At1g20620), as well as the cDNAs of *APX1* (1270 bp, At1g07890) and *Cu/ZnSOD1* (873 bp, At1g08830), were cloned together with the cDNA of *WRKY53* (1514 bp, At4g23810) into the pBiFCt-2in1-NN vector carrying both genes of interest on one vector. In addition, an internal red fluorescent protein (*RFP*) gene as transformation and expression control was localized on the same vector backbone. The genes of interest were fused to the N- or the C-terminal part of the yellow fluorescent protein (*YFP*), respectively. The expression of the fusion proteins is driven by the cauliflower mosaic virus 35S promoter [34]. For the transfection of the protoplasts, 4 µg of the plasmid DNA was used to gain expression of the fusion proteins. If the fusion proteins can interact with each other, yellow fluorescence is restored by bringing the YFP-N and YFP-C parts into close proximity. These interactions were visualized by flow cytometry using CytoFLEX (Beckman Coulter, Brea, CA, USA) 1 day after transfection. The internal RFP and any reconstituted YFP were both excited by the onboard 488 nm laser. Emissions were captured for YFP in FL1 (525/40 nm) and RFP in FL3 (610/20 nm), respectively. All interaction tests were performed at least 3 times independently. In addition, confocal microscopy (LSM880, Zeiss, Jena, Germany) of transfected tobacco leaves was used to detect and localize the interaction within the cells. For this purpose, *N. benthamiana* plants were infiltrated with *A. tumefaciens* suspension cultures containing the pBiFCt-2in1 constructs described above. Microscopic analyses were performed 2 days after infiltration. At least 3 leaves of different plants were analyzed under a Zeiss LSM 880 Airyscan confocal microscope by using the preset sequential scan settings for YFP (Ex: 514 nm, Em: 517–553 nm) and for RFP (Ex: 561 nm, Em: 597–625 nm). The experiments were repeated at least 3 times.

### 2.5. Intracellular Hydrogen Peroxide Measurements

Whole rosette leaves were harvested from the same positions within the rosette and then incubated in 9.5 µM 5(6)-Carboxy-Di-Hydro-Di-Chloro-Fluorescein-Di-Acetate (Carboxy-H_2_DCFDA) in MS-Medium (pH 5.7). Samples were incubated for 45 min in the dye solution, washed twice with distilled water, then frozen in liquid nitrogen. Samples were homogenized on ice in 500 µL 40 mM Tris-HCl pH 7.0. After centrifugation (30 min, 4 °C, 14,000 rpm), fluorescence of supernatant was measured (480 nm excitation, 525 nm emission, Berthold TriStar LB941, BERTHOLD TECHNOLOGIES, Bad Wildbad, Germany). The H_2_DCFDA solution needs to be calibrated by chemical de-acetylation and oxidation following, and it has to be prepared freshly for each sample harvest [35].

### 2.6. Gene Expression Analyses Using qRT-PCR

The total RNA was extracted with the GeneMATRIX Universal RNA Purification Kit (*EURx*, Gdańsk, Poland). RevertAid Reverse Transcriptase (Thermo Fisher Scientific Inc., Waltham, MA, USA) and oligo-dT primers were applied for cDNA synthesis. KAPA SYBR^®^ Fast Bio Rad iCycler (Bio-Rad Laboratories Inc., Hercules, CA, USA) and Master Mix were used for qRT-PCR analyses following the manufacturer’s protocol. We used the ∆∆CT method for calculation according to [36]. *WRKY53* expression was normalized to *ACTIN2* values and given in % of *ACTIN2*. Primer design for qRT-PCR was performed via QuantPrime [37], *ACTIN2* (At3g18780): AAGCTCTCCTTTGTTGCTGTT and GTTGTCTCGTGGATTCCAGCAGCTT, WRKY53 (At4g23810): CAGACGGGGATGCTACGG and GGCGAGGCTAATGGTGGT.

### 2.7. Purification of 8xHis-Tagged WRKY Proteins

WRKY proteins were ordered as N-terminally 8xHis-tagged proteins from Biomatik (Cambridge, ON, Canada). Proteins were expressed in *E. coli* cells and purified by affinity purification. Concentrations of WRKYs were determined by Bradford protein measurements, while quality and purification were shown by SDS-PAGE, Coomassie staining and Western blotting, followed by immune detection using anti-HIS antibodies (Appendix A).

### 2.8. GUS-Reporter Gene Assays

Transiently transformed *Arabidopsis* leaves were incubated overnight in a staining solution containing a 100 mM sodium phosphate buffer pH 7.5, 10 mM EDTA pH 8.0, 0.5 mM potassium ferricyanide (K_3_[Fe(CN)_6_]), 0.5 mM potassium ferrocyanide (K_4_[Fe(CN)_6_]), 0.1% Triton X-100 and 0.5 mg/mL X-GlcA (Cyclohexylammonium salt, Duchefa, Haarlem, The Netherlands). Subsequently, the chlorophyll had to be removed for better analysis of the blue staining. This was achieved by shaking the plant material in 80% ethanol for another 24 h, during which the ethanol solution was changed several times.

### 2.9. Antioxidative Capacity of Leaf Discs

The decomposition of H_2_O_2_ can be evaluated using commercially available peroxide strips (Dosatest peroxide test strips 100, VWR Chemicals, Leuven, Belgium). Therefore, we excelled leaf discs from wild-type or transiently transformed *N. benthamiana* leaves, and these discs were incubated in a 1 mM H_2_O_2_ solution. To measure the H_2_O_2_ decomposition activity of these discs, one of the test strips was submerged for 1 s into the solution immediately after placing the leaf disc into the solution as baseline (time point 0 min). This procedure was repeated after 2 h. The given control color scale can be used to read out the amount of residual peroxide, and the weaker the blue color, the less peroxide is present in the solution.

## 3. Results

*WRKY53* gene expression, activity and degradation are tightly controlled during plant devolvement and senescence. The WRKY53 protein can interact directly with a variety of partners in different cellular compartments ([14] and references therein). According to different yeast two-hybrid screens, WRKY53 appears to have many more different interaction partners. In order to analyze the actual in vivo protein-interaction partners of WRKY53 over plant development, we performed Co-IP experiments with 35S:*WRKY53* overexpressing plants in *upl5* background using anti-WRKY53 antibodies at different time points, then analyzed the pulled-down proteins with LC-MS/MS. To our surprise, we found many antioxidative enzymes among the direct interaction partners of WRKY53, including different isoforms of CATs, APXs and SODs (see Appendix A).

### 3.1. Inhibition of the Activity of Different Antioxidative Enzymes by WRKY53

In order to evaluate whether this interaction with WRKY53 has any consequences for the function of the antioxidative enzymes, we analyzed the activity of the different isoforms of CATs, APXs and SODs using zymograms in the presence of WRKY53 protein. Tagged versions of WRKY53, as well as of WRKY18 and WRKY25, were expressed in *E. coli* cells and purified via the 8xhis tag. The WRKY53 highly enriched protein factions (Appendix A) were added in increasing amounts to the crude plant protein extracts isolated from wild-type Col-0 and *wrky53* mutant plants, and myelin basic protein (MBP) was used as control protein. After incubation, the influence of WRKY53 and MBP on the activity of the different isoforms of the antioxidative enzymes was tested by enzyme-specific zymograms. Addition of WRKY53 protein had a clear inhibiting effect on the activity of the CAT2 and CAT3 homotetramers, as well as the activity of the heterotetramers (Figure 1A). The catalase isoforms can be clearly differentiated by their different reactions towards 3’amino 1,2,4 triazole, or by using catalase single and double mutants [21,30]. Inhibition appears to be concentration-dependent, and the active protein complexes appear to be slightly shifted upwards with increasing amounts of added proteins. In contrast, MBP had no effect on the activity of CAT2, CAT3 or the heterotetrameres. Moreover, the addition of WRKY18 highly enriched protein factions, as well as of WRKY25 highly enriched protein factions, had also no effect on CAT activities, indicating a WRKY53 specific inhibition (Figure 1A). As expected, CAT2 activity is higher in leaf tissue compared to CAT3; however, the protein amount appears to be more equal according to Western blotting of a native protein gel and subsequent immune detection with anti-rye catalase antibodies (Appendix A).

Even though eight genes and several isoforms of ascorbate peroxidase exist, it is difficult to see their activities on the zymograms of crude extracts. The most prominent visible isoform is the cytosolic APX1, which can only be visualized if high amounts of ascorbate are present during the extraction and electrophoresis procedure to stabilize the enzyme. Again, the addition of WRKY53 to the crude protein extracts inhibits enzyme activity of APX1 in a concentration-dependent manner (Figure 1B). As for the CATs, the addition of MBP, as well as of WRKY18, had no effect. In this case, the addition of WRKY25 could slightly inhibit APX1 activity. Furthermore, SOD activities were inhibited by WRKY53; however, here the different isoforms were inhibited to different extents. The Cu/ZnSOD activities declined most prominently, followed by the FeSOD, while the Mn-SOD appeared to be insensitive (Figure 1C). As for APX1, the addition of MBP, as well as of WRKY18, had no effect and, again, addition of WRKY25 protein could slightly inhibit the Cu/ZnSODs.

In summary, the presence of WRKY53 selectively inhibits the activity of specific isoforms of the antioxidative enzymes, and WRKY53 is more effective than other WRKYs.

### 3.2. Protein–Protein Interaction between WRKY53 and Different Antioxidative Enzymes

To confirm that the interaction between WRKY53 and CAT2, CAT3, APX1 and CuZn-SOD1 can also be observed in living cells, we used Bimolecular Fluorescence Complementation (BiFC) assays in transiently transformed *Arabidopsis* protoplasts and *N. benthamiana* leaves. Therefore, WRKY53 was fused with the C-terminal half of YFP, while the potential interaction partners were combined with the N-terminal half of YFP, or vice versa. If interaction takes place, the two halves of the YFP come into close proximity and are able to emit a yellow fluorescence. In the case of the transiently transformed Arabidopsis protoplasts, fluorescent cells were sorted in a CytoFLEX cell sorter, in which a portion of the cells showed a YFP fluorescence, clearly indicating in vivo interaction. These experiments clearly confirmed that CAT2, CAT3, APX1 and Cu/ZnSOD1 proteins can directly interact with the transcription factor WRKY53 (Figure 2A, Appendix A). In addition, these interactions were analyzed in transiently transformed tobacco leaves using confocal microscopy. Again, yellow fluorescence can only be emitted if the two proteins interact and bring the two halves of the fluorescent protein together. Not only do the microscopy pictures confirm the interactions, the intracellular localization of the interaction could also be observed under the microscope. CAT2 and CAT3 could form a complex with WRKY53 in the peroxisomes, as well as in the nucleus; this means that either WRKY53 is taken to the peroxisomes via the interaction with the CATs, or CAT enzyme complexes or their subunits are translocated to the nucleus by the interaction with WRKY53 (Figure 2B). Interaction between WRKY53 and APX1 or Cu/ZnSOD1 has predominantly been observed in the nucleus; however, the cytoplasmic signal might have been too low and dispersed to be detected (Figure 2B). Again, so far, no nuclear localization of APX1 or Cu/ZnSOD1 was reported, so we can conclude that APX1 and Cu/ZnSOD1 are also translocated to the nucleus via the interaction with WRKY53, as was already described for another protein [14]. Yet, in contrast to catalase subunits (57 kDa), both proteins are small in size (27.5 kDa and 16 kDa, respectively), so that, in principle, they could also diffuse freely between the nucleus and cytoplasm [38].

In addition, the antioxidative capacity of these transiently transformed tobacco leaves was tested using leaf discs, which were then incubated in 1 mM H_2_O_2_ solution. We could again confirm that the presence of WRKY53 lowers the overall antioxidative capacity towards H_2_O_2_, even if CAT2, CAT3, APX1 or SOD1 were co-expressed (Figure 3). The leaf discs of wild-type *N. benthamiana* were able to detoxify almost all peroxides in a 1 mM solution within 2 h so that only approx. 1 mg/L peroxide was left after the incubation. However, if *WRKY53* was expressed in the leaf tissue, the antioxidative capacity of the leaf discs was lower, and still between 3–10 mg/L peroxide were left after 2 h. Moreover, if *CAT2*, *CAT3* or *APX1* were co-expressed, a higher antioxidative capacity towards H_2_O_2_ would have been expected, which was not the case. By overexpression of SOD1, if at all, an additional production of H_2_O_2_ would have been expected; however, in all cases the presence of WRKY53 appeared to block the function of the antioxidative enzymes.

### 3.3. Inhibition of Antioxidative Enzymes by WRKY53 during Plant Development and Onset of Senescence

WRKY53 was shown to be one important regulatory hub in the complex network of senescence regulation. *WRKY53* gene expression, activity and degradation are tightly regulated, including even several double bottoms, e.g., *WRKY53* is highly expressed at the onset of monocarpic senescence in approx. 6- to 7-week-old plants while specific degradation of the WRKY53 protein by the HECT domain E3 ubiquitin ligase protein UPL5 is diminished at the same time by downregulation of *UPL5* expression [16,26]. Moreover, *WRKY53* gene expression can be induced by hydrogen peroxide, which increases during bolting and flowering time at the onset of monocarpic senescence [21,26]. The downregulation of *CAT2* expression, combined with an increased sensitivity of APX1 activity against hydrogen peroxide at bolting and flowering time, appear to be responsible for the production of this peak [21,22,24]. Therefore, we wanted to analyze whether inhibition of the antioxidative enzymes CAT and APX is possible throughout development and whether activity profiles change over development in a *wrky53* mutant plant. As already observed before, CAT2 activity decreased while CAT3 activity increased with age (Figure 4A). When WRKY53 protein was added to these different extracts, CAT2 and CAT3 activity could be inhibited, and an upwards-shift of the protein complexes could be observed in all developmental stages (Figure 4A). APX1 activity was down-regulated during bolting time but recovered at later stages (Figure 4B) [21,24]; this downregulation coincided with the increase in intracellular hydrogen peroxide and the expression of *WRKY53* (Figure 4C,D). Remarkably, the increase in intracellular hydrogen peroxide was more pronounced in younger than in older leaves, emphasizing the signaling character of hydrogen peroxide. Again, the addition of WRKY53 protein to the extracts could inhibit the activity of APX1 in all stages. For the SODs, we could not detect any obvious activity changes over development, and so this was not further analyzed.

In *wrky53* mutant plants, the down-regulation of CAT2 activity is diminished and/or delayed while the increase of CAT3 activity is accelerated, indicating that the WRKY53 protein contributes to the regulation of the activity of both catalases during development (Figure 5A). Still, the addition of WRKY53 protein to the extracts of *wrky53* mutants can inhibit both isoforms (Figure 5A). For APX, the decrease of APX1 activity during bolting and flowering time is prolonged in *wrky53* mutant plants (Figure 5B), which is consistent with the delayed senescence phenotype of these plants [24,26]. Again, the addition of WRKY53 protein to these extracts could inhibit APX1 activity severely (Figure 5B).

### 3.4. Influence of the Complex Formation with the Antioxidative Enzymes on WRKY53 Function

As the protein interaction can lead to the inhibition and/or inactivation of the antioxidative enzyme, we wanted to know which influence this protein interaction has on the function of WRKY53. Therefore, we used reporter gene assays, in which a promoter sequence that can bind WRKY53 is driving the expression of the reporter gene ß-glucuronidase (*GUS*). In this reporter system, we used the promoter sequence of *WRKY53* itself (a 2,759-bp-sequence upstream of the start codon) and a 35S-driven *WRKY53* construct as effector together with 35S-driven constructs of *CAT2*, *CAT3*, *APX1* and *Cu/Zn*-*SOD1* as co-effectors. As observed before, WRKY53 regulates its own expression in a negative feedback loop, which is demonstrated by the lower *GUS* reporter gene expression and enzyme activity in transiently transformed *Arabidopsis* leaves (Figure 6A,B) [26]. One example of the histochemical GUS staining of these leaves is presented in Figure 6A, and a quantification of the stained regions of several transformed leaves is shown in Figure 6B,C. When we added 35S:*CAT2* or 35S:*SOD1* effector constructs as co-effector to the 35S:*WRKY53* and the P*_WRKY53_*:*GUS* reporter, the negative effect of WRKY53 on the *GUS* expression was suppressed, indicating that CAT2 and Cu/Zn-SOD1 interactions with WRKY53 do not only inhibit enzyme activities of CAT and SOD but also block WRKY53 function as transcriptional repressor of its own promoter. APX1 and CAT3 interaction with WRKY53 also appear to slightly inhibit WRKY53 function, but here the observed differences were not statistically significant.

## 4. Discussion

It has become evident during the last two decades that ROS, especially hydrogen peroxide, can have a signaling function in developmental as well as stress-induced senescence. However, the mechanisms by which plants sense this parameter, and how specificity can be achieved, is still not well understood. So far it has been made clear that the ROS concentrations in the cytoplasm and different organelles have a different impact on senescence and are tightly regulated. Consistently, CAT and APX isoforms in particular are positioned strategically in different cellular compartments to control H_2_O_2_ levels and to ensure their function as signaling molecules [39]. Interestingly, H_2_O_2_ from chloroplasts and peroxisomes modulates the plant transcriptome differentially and has different impacts on senescence [40,41]. Furthermore, many senescence-associated transcription factors are upregulated in their own expression by H_2_O_2_ [3,17,26] or influenced in their activity by the redox conditions in the cytoplasm and nucleus [14,17,19,20] Overall, a very complex regulatory network has to be in place to regulate senescence and, more specifically, intracellular H_2_O_2_ concentrations.

Obviously, the complex regulatory network of senescence comprises many regulatory feedback loops, including the direct binding of *CAT1*, *CAT2*, and *CAT3* promoter elements by WRKY53, which activates the expression of the catalases, leading to the reduction of the intracellular hydrogen peroxide levels which, in turn, reduce expression of the *WRKY53* gene. Consistently, *cat2/3* double mutants with higher intracellular H_2_O_2_ levels showed increased expression of many WRKYs, including *WRKY53* [42]. In contrast, WRKY75 negatively regulates the expression of *CAT2* and *SID2*, leading to increasing salicylic acid and H_2_O_2_ levels, and thereby to a gradual but self-sustained rise of *WRKY75* expression during senescence driven by three interlinking positive feedback loops [43]. Moreover, WRKY53 can induce the expression of its family member *WRKY25* and, vice versa, WRKY25 can upregulate the expression of the *WRKY53* gene, creating a positive feedback regulation. At the same time, *WRKY25* expression diminishes intracellular H_2_O_2_ contents, which, in turn, leads to decreased *WRKY53* expression. As for *WRKY53*, expression of *WRKY25* can be induced by H_2_O_2_, and WRKY53 is involved in this induction. In addition, WRKY25 negatively regulates its own expression, which might prevent an overshooting of the reaction to H_2_O_2_ [17]. At first glance, these feedback regulations appear to be complex but contradictory at some points; however, these are only parts of a larger regulatory network which is likely to contain many more feedback controls. Here, we could add another feedback mechanism to the circuit between WRKYs, H_2_O_2_ and antioxidative enzymes. The WRKY53 protein can directly interact with different antioxidative enzymes, including CATs, APXs and even SODs (Figure 2, Appendix A) and inhibit their function. This inhibition appears to be selective for specific isoforms, as well as for specific WRKY factors (Figure 1) in which WRKY53 is more effective than WRKY25, whereas WRKY18 had no effect. On the other hand, *WRKY53* expression is upregulated in a *cat2/3* mutant more prominent than *WRKY25* or *WRKY18* [42], also indicating that WRKY53 plays a more central role in this feedback regulation. Moreover, the inhibition of CATs and APXs by WRKY53 can take place throughout a wide range of different developmental stages, as the activity profiles differ between Col-0 and *wrky53* mutant plants (Figure 4 and Figure 5). It was shown before that inhibition of APX activity during bolting is executed post-transcriptionally [21,44,45]. However, as this inhibition of APX activity during bolting is still visible in *wrky53* mutant plants, WRKY53 interaction appears to not be the main responsible mechanism here, even though *WRKY53* expression level rises exactly at this time point. Moreover, the inhibition of APX1 appears to be prolonged in the *wrky53* mutants, indicating that this prolonged inhibition is due to the delayed senescence in these plants [26].

Further to WRKY53, CATs can interact with many different non-peroxisomal proteins including cytoplasmic and nuclear proteins, nicely reviewed in [39]. This is possible, as CATs are not exclusively located in the peroxisomes and can also be found in the cytoplasm [46]. It has been shown before that CATs can be retained in the cytosol under oxidative stress conditions in which the peroxisomal import receptor PEX5 functions as a stress sensor [47]. Interaction of CAT3 with a cucumber mosaic virus protein 2b can translocate CAT3 to the nucleus [48]. Here, we could show that the interaction between WRKY53 and CAT2 or CAT3 can direct the resulting complexes to the peroxisomes but also appears to foster the translocation of the CATs to the nucleus (Figure 2). The interaction inhibits catalase activity, but whether CAT is released again from the WRKY53 interaction in the nucleus is not yet clear. Moreover, whether the whole catalase complex, which consists of four subunits and four HEME groups, or only the protein subunits are imported to the nucleus has still to be analyzed. However, the interaction with the oxidoreductase NUCLOREDOXIN1 protects CATs from ROS-induced oxidation and is required for their optimal function under oxidative stress conditions [49], indicating that the interaction with the nuclear protein NUCLOREDOXIN1 involves the functional tetramer.

Furthermore, CAT activity can also be regulated by other interactions: CAT3 can interact specifically with CALCIUM-DEPENDENT PROTEIN KINASE8 and can be phosphorylated at Ser-261, which positively influences its activity [46]. In addition, CAT3 can directly interact with S-nitrosoglutathione reductase (GSNOR) and acts in this case as a transnitrosylase that specifically modifies GSNOR1 at Cys-10, which CAT2 or CAT1 cannot, indicating that CAT3 is also involved in NO redox signaling in plants [50]. This *S*-nitrosylation of GSNOR1 induces a conformational rearrangement and fosters AUTOPHAGY8 binding, which then can promote its degradation via autophagy [51]. Whether WRKY53 can also be transnitrosylated by CAT3 interaction will be the subject of further investigations.

Taken together, we provide evidence for a new feedback regulation between the transcription factor WRKY53 and the antioxidative enzymes, which is illustrated for CAT2 in a model (Appendix A).

## 5. Conclusions

Senescence is an important developmental process that substantially contributes to the fitness of plants. The aim of senescence is to optimize and efficiently utilize carbon, nitrogen, and mineral resources. Reallocation of these resources from senescing tissues to maturing seeds or fruits has also a major impact on yield quantity and the quality of crop plants. However, to maximize and guarantee quality and quantity of crop harvests, the correct timing of onset and progression is essential. In contrast, premature senescence, which can be induced by abiotic and biotic stresses, is often responsible for crop losses. Therefore, understanding the regulatory networks coordinating this process will in the long-run be helpful for tightly controlling senescence and avoiding premature induction. In contrast to other research groups, which take more systemic approaches, we aim to decipher the complex regulatory network around the transcription factor WRKY53 in the model plant *Arabidopsis* in more detail. We have already uncovered different regulatory mechanisms to control expression, activity and protein amount of this transcription factor. Here, we found a new and additional regulatory cue in which a complex formation between different antioxidative enzymes and the WRKY53 protein leads to the inactivation of both partners. However, as H_2_O_2_ can boost the expression of the *WRKY53* gene, this creates a new positive feedback loop, as inhibition of CAT and APX activity will lead to an increase in H_2_O_2_ content, which further activates *WRKY53* expression. However, at least in the case of the catalases, increasing amounts of WRKY53 will activate the gene expression of all three catalases, which, in turn, then reduces H_2_O_2_ contents, again creating a negative feedback loop. Moreover, the responsiveness of *WRKY53* expression to H_2_O_2_ is dependent on the developmental stage. This indicates that a delicate balance between H_2_O_2,_ WRKY53 protein amount, antioxidative enzyme amount and activity is installed in the leaf cells of *Arabidopsis*. As catalases in particular have a plethora of different interaction partners, the antioxidative enzyme might have a role in coordinating many different cellular functions. In the future, this knowledge will be fed into modelling approaches to achieve better insights into the functioning of such complex regulatory networks [52], which might enable us to simulate what happens if specific components of the network are manipulated. In this sense, a detailed understanding of senescence regulation in the model plant *Arabidopsis* will provide new candidates for the further improvement of agricultural plants.

## Figures and Tables

**Figure 1 antioxidants-13-00315-f001:**
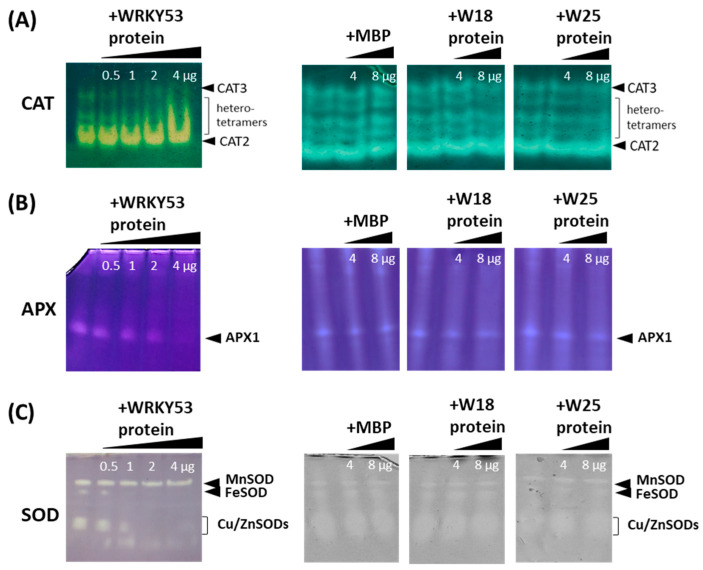
Zymograms of crude protein extracts isolated from leaf tissue of wild-type *Arabidopsis* plants. Staining for (**A**) CAT activity, (**B**) APX activity, (**C**) SOD activity. Visible isoforms are indicated by the arrows. Increasing amounts of highly enriched protein factions of WRKY53 (0.5–4 µg) or WRKY18 (4, 8 µg), WRKY25 (4, 8 µg), were added to the extracts before loading. MBP (4, 8 µg) was used as control protein, as indicated on each gel and lane.

**Figure 2 antioxidants-13-00315-f002:**
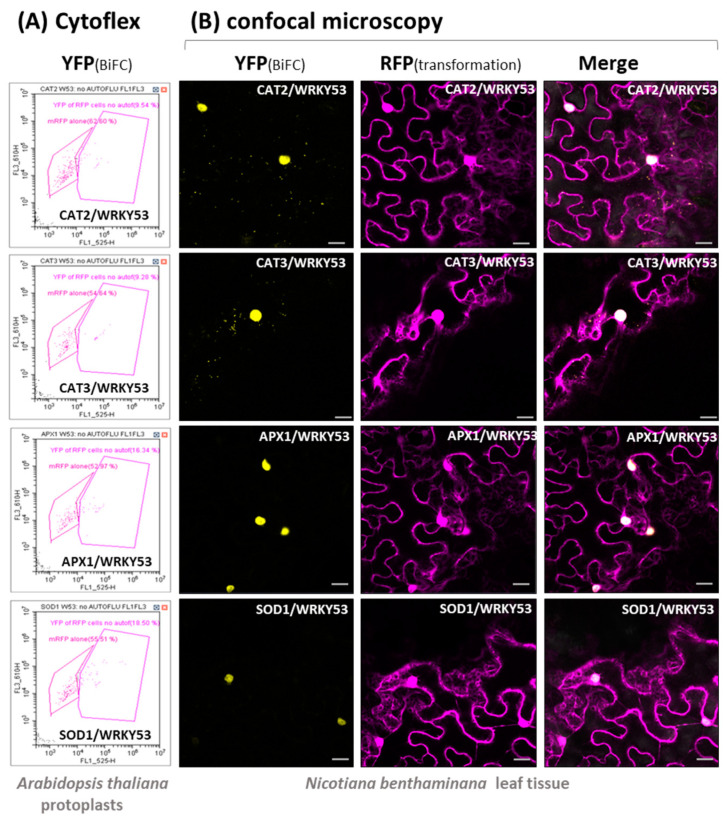
In planta protein–protein interaction between WRKY53 and CAT2, CAT3, APX1 or Cu/ZnSOD1 using BiFC in transiently transformed *Arabidopsis* protoplasts or tobacco leaves. (**A**) Protoplasts were analyzed with the CytoFLEX cell sorter: pink framed polygons indicate transformed protoplasts (RFP fluorescence, left polygon), purple framed polygons indicate interaction via BiFC (YFP fluorescence, right polygon). (**B**) Transiently transformed *N. benthamiana* leaves were analyzed under the confocal laser scanning microscope: yellow fluorescence (YFP) indicates BiFC, red fluorescence (RFP) is used as a transformation control. Scale bar indicates 20 µm.

**Figure 3 antioxidants-13-00315-f003:**
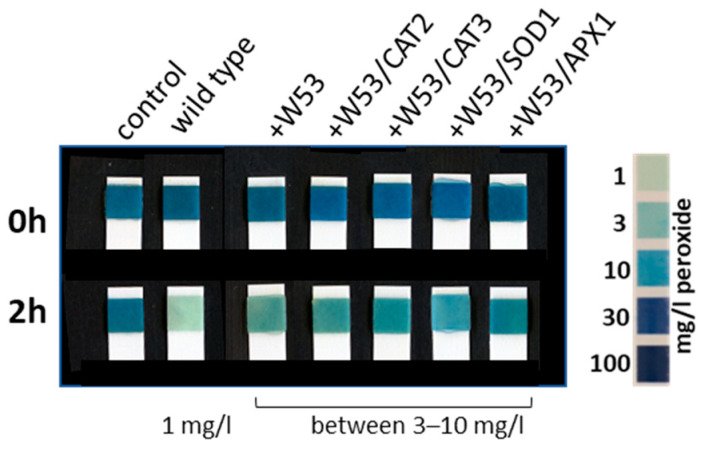
Leaf discs of wild-type or transiently transformed *N. benthamiana* leaves were incubated in a 1 mM H_2_O_2_ solution. At time points 0 h and 2 h, the concentration of H_2_O_2_ was determined using commercially available peroxide stripes. Color scale for peroxide content is provided on the right. Transformed effector constructs are mentioned above the stripes. As control, H_2_O_2_ solution without leaf discs was measured (left).

**Figure 4 antioxidants-13-00315-f004:**
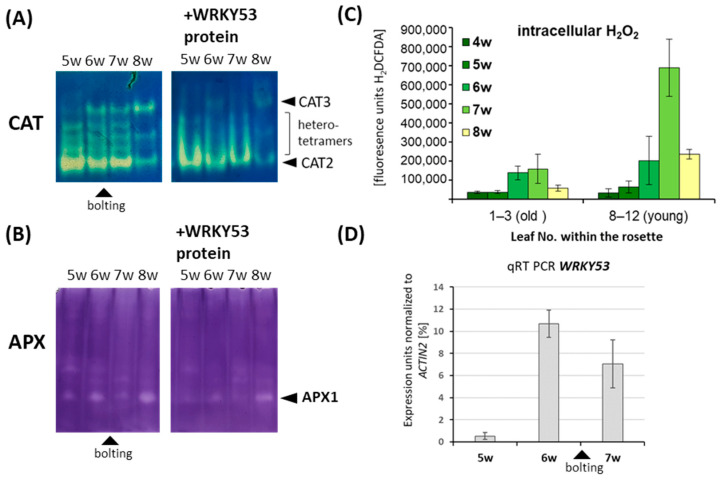
Zymograms of crude protein extracts isolated from leaf tissue of wild-type *Arabidopsis* plants at different developmental stages from 5 to 8 weeks (w). Staining for (**A**) CAT activity, (**B**) APX activity, 4 µg of a highly enriched protein factions of WRKY53 were added to the extracts before loading. Visible isoforms and bolting time are indicated by the black arrow heads. (**C**) H_2_O_2_ content was measured using H_2_DCFDA fluorescence in pools of young (No. 8–12) and old leaves (No. 1–3) of the same rosette which were harvested from 4- to 8-week-old plants. H_2_O_2_ contents are indicated in arbitrary units of H_2_DCFDA fluorescence, and error bars indicate ±SD, *n* = 3. (**D**) WRKY53 expression was analyzed by qRT-PCR in pools of rosette leaves No. 6 and 7 of three plants and were normalized to *ACTIN2* expression. Error bars indicated ±SE, *n* = 3.

**Figure 5 antioxidants-13-00315-f005:**
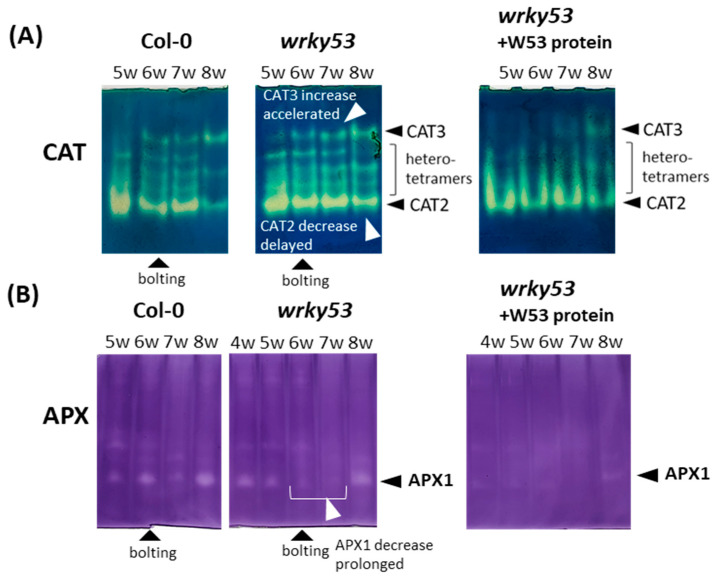
Zymograms of crude protein extracts isolated from leaf tissue of wild-type *Arabidopsis* (Col-0) or *wrky53* mutant plants at different developmental stages from 5 to 8 weeks (w). Staining for (**A**) CAT activity, (**B**) APX activity, 4 µg of a highly enriched protein factions of WRKY53 (W53) were added to the extracts before loading. Visible isoforms and bolting time are indicated by the black arrow heads. White arrowheads point to changes in the enzyme activities between Col-0 and *wrky53*.

**Figure 6 antioxidants-13-00315-f006:**
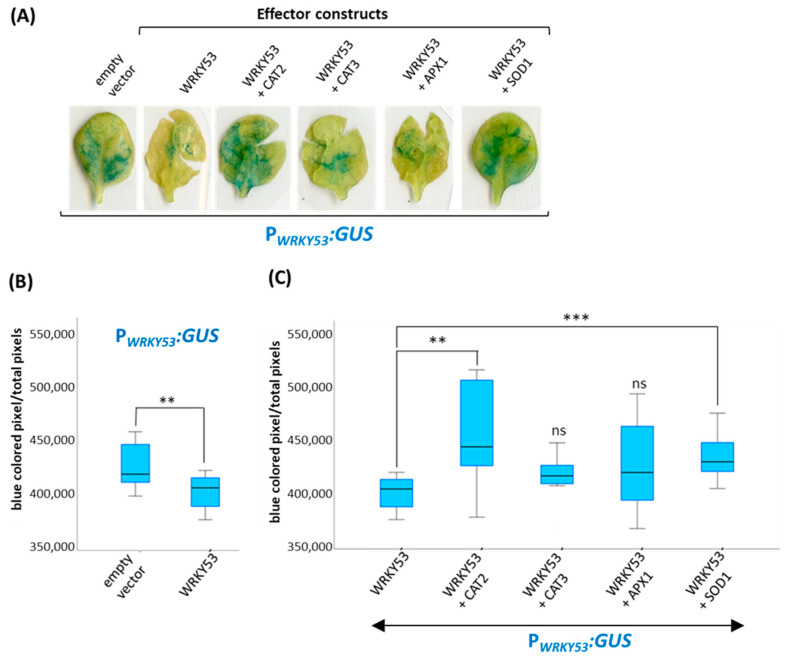
Histochemical GUS staining of transiently transformed *Arabidopsis* leaves. All leaves were transformed with a P*_WRKY53_*:*GUS* construct. Both 35S-driven effector and co-effector constructs are indicated, while the empty vector was used as control. (**A**) One example of the GUS stained leaves is presented. (**B**) Effect of WRKY53 on the expression of its own promoter compared to the empty vector control, a boxplot of the quantification of the GUS staining is presented. (**C**) Effect of WRKY53 in the presence of different antioxidative enzymes as co-effectors, and a boxplot of the quantification of the GUS staining is presented. At least eight transformed leaves were analyzed, respectively. A *t*-test was performed for significant differences (*n* = 8–13), ** *p* ≤ 0.01, *** *p* ≤ 0.001, ns = not significant.

## Data Availability

The original contributions presented in the study are included in the article and Appendix A, further inquiries can be directed to the corresponding author.

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
