# Peer review of "Complex Formation between the Transcription Factor WRKY53 and Antioxidative Enzymes Leads to Reciprocal Inhibition"

_antioxidants, 2024, doi:10.3390/antiox13030315_

Round 1
Reviewer 1 Report
Comments and Suggestions for Authors
The authors identified different antioxidative enzymes including catalases (CATs), superoxide dismutases (SODs) and ascorbate peroxidases (APXs) as protein interaction partners of WRKY53 in a experiment at different developmental stages.
The reviewer want to know: gene WRKY family is large gene family. How do you identified them and major in WRKY53? Has the author done the basic location of the gene WRKY53? Whether RT-qPCR has been used to detect gene expression in different tissues?
In Figure 1, 3 and 5, it is suggested that the authors should use histograms to represent the activities of CAT, APX and SOD, other enzymes.
Line 92, italics
Line 129, FeCl3, K3(FeCN)6;
Line 163, 185, When Latin first appears, it is the full name, and when it appears the second time, it is abbreviated;
Line175, Lower case
Line 181, 488 nm

English is good with a few errors.
Author Response
Please see the attachement

Reviewer 2 Report
Comments and Suggestions for Authors
The researchers evaluated the activity of different antioxidant enzymes, including catalases (CATs), superoxide dismutases (SODs) and ascorbate peroxidases (APXs) as protein interaction partners of WRKY53 at different developmental stages of Arabidopsis thaliana. The study is interesting because it studies a new metabolic pathway linked to the WRKY53 transcription factor. However, there is a need to review the manuscript to improve its scientific quality. The authors need to justify why they used the Arabidopsis thaliana model and possibly the knowledge generated will not benefit cultivated species with agronomic importance. There are other models in cultivated species such as tomato (Micro-tom).
It is known that several micronutrients constitute the co-factor of the antioxidant enzymes studied and can affect their activity, which can greatly decrease in the case of nutritional deficiency and consequently alter the results of the study. Therefore, the authors did not indicate whether the plants had the appropriate nutritional status for all nutrients. They did not indicate the amount of nutrients provided in the soil and the details of the soil used. Plant water management is also important because it affects nutrient absorption. The authors did not indicate the specific leaf that was collected from the plant for each sampling carried out during the 4 to 8 week period. The age of the leaf directly affects the level of phloem redistribution of nutrients and consequently the results. Even in each collection of plants, it is interesting to reinforce the phenological stage of the plant.
A multivariate analysis of the data may allow for a deeper discussion of the results obtained. It is important for the authors to indicate at the end of the work the practical implications of the discovery.
Author Response
Please see the attachement

Reviewer 3 Report
1. WRKY53 is a transcription factor, and considering that catalase can also translocate to the nucleus, the interaction of WRKY53 with antioxidative proteins in this study suggests a direct interaction, rather than regulatory action via promoter binding. The mechanism behind this interaction should be explored through literature research and included in the “Discussion” section.
1. In the Abstract section, Lines 20-22, 24-25, the statements “The interaction with WRKY53 inhibited the activity of different enzyme isoforms to varying degrees, and conversely, the function of WRKY53 was also inhibited by these complex formations.” and “We identified a novel and unexpected feedback regulation mechanism involving H2O2, antioxidative enzymes, and the transcription factor WRKY53.” are somewhat vague. This vagueness is partly attributed to the non-specificity of the research findings. Given the extensive research on the relationship between WRKY53 and antioxidative enzymes, this study should strive for greater specificity and enhance the robustness of its results with support from other literature. A notable instance of this lack of specificity in the manuscript is observed in Lines 56-58: “The HECT ubiquitin ligase (UPL5), which exhibits an expression pattern opposite to that of WRKY53, ensures control of the protein content, even if mis-expressed.” This statement should be clarified: is it referring to a process governed by WRKY53 degradation, or the elimination of misexpressed proteins?
2. The use of “long term” in the context of hydrogen peroxide production lacks precision. The specific timeframe referred to should be defined.
3. The claim that “WRKY53 selectively inhibits the activity of specific isoforms of antioxidative enzymes and is more effective than other WRKY family members” lacks clarity without more precise quantitative data on isoform changes. Furthermore, to support claims of isoform variation, additional corroborative experimental data should be presented.
4. What does it imply when WRKY53 interacts with antioxidative proteins? If it implies direct binding, what is the regulatory mechanism involved? Being a transcription factor, does WRKY53 directly bind to antioxidative proteins in peroxisomes and inhibit their enzymatic activities?
5. The methodology for differentiating isoforms should be elucidated. Specifically, the rationale behind distinguishing CAT2 and CAT3 in zymograms needs clarification, even if it involves citing other literature.
6. The “Model of feedback regulation between WRKY53 and CAT2” cannot be considered a representation of the findings from this study. The study's findings or the sources of cited materials should be clearly indicated.
7. Minor points:
1. Line 131: Clarify the term "PAA gels".
2. Line 131: Correct the subscript notation for FeCl3 and K3[Fe(CN)6].
Author Response
please see the attachement

Round 2
Reviewer 1 Report
The authors should pay attention for some bisical punctuation and Latin italics.
All Latin names need italics;
Line 516,706 with a spaceï¼›
Line 712,737,777, add “‒” between the two numbers;

Author Response
Thank you for your helpful comments, which contributed to imporve the manuscript
All Latin names need italics;
This has been changed throughout the manuscript
Line 516,706 with a spaceï¼›
Has been changed
Line 712,737,777, add “‒” between the two numbers;
Has also been changed
Reviewer 2 Report
The authors revised the manuscript and improved its quality and we recommend its approval.
The authors revised the manuscript and improved its quality and we recommend its approval.
Author Response
We have improved the mansucript according to your advice. Thank you for the helpful comments.